# Personalized Federated Learning With Similarity Information Supervisor

## Abstract

A crucial issue in federated learning is the heterogeneity of data between clients, which can lead to model weight divergence, eventually deteriorating the model performance. Personalized federated learning (pFL) has been proven to be an effective approach to addressing data heterogeneity in federated learning. However, existing pFL studies seldom verify whether the broadcast global model is beneficial for the local model performance. To address this, we propose a novel pFL method, called federated learning with similarity information supervision (Fed-SimSup). Specifically, FedSimSup incorporates a local supervisor to assist the model training and a personalized model for global information aggregation. The role of the supervisor is to refine the personalized model when it is not beneficial for the local model performance, ensuring the effective global information aggregation while aligning with the local heterogeneous data. Additionally, the similarity relationships between the clients are measured using label distribution differences of the local raw data to weight the personalized models, promoting information usage among similar clients. Experimental results demonstrate three advantages of FedSimSup: (1) It shows better performance over heterogeneous data compared with seven state-of-the-art federated learning methods; (2) It can allow for different model architectures across different clients; (3) It offers a certain degree of interpretability.

## 1 Introduction

In the digital age, data privacy has become increasingly important, which stands in contrast to the growing demand for data in artificial intelligence. In response to the challenges of data privacy, federated learning (FL) has experienced rapid growth (Cheng et al., 2020). The goal of FL is to maximize the utilization of each client's data while preserving their privacy and minimizing communication costs, by training a comprehensive global machine learning model. In typical FL, the overall process is as follows: (1) participating clients first download the latest model from the server for local use. (2) clients train and update the model on their local datasets. (3) clients upload the updated model to the server. (4) the server then aggregates the models collected from multiple clients and updates the global model, which is provided to clients involved in subsequent communications.

When dealing with independent and identically distributed (IID) data, the most popular FL method FedAvg (McMahan et al., 2017) is guaranteed to converge and delivers good performance. However, in real-world scenarios, Non-IID data is more common, and this heterogeneous setting will slow down the convergence and degrade the learning performance (Zhao et al., 2018). To address this, in recent years, personalized federated learning (pFL) (Tan et al., 2022a) has been developed as one of the effective methods to address challenges caused by the Non-IID data. Mainstream pFL can be categorized into two primary directions. One approach focuses on training a more robust global model that can generalize effectively across all clients. The other approach is to train personalized models for each client to address the issue of data heterogeneity. The two directions address the Non-IID problem to some extent from different perspectives.

From the perspective of an individual client in the FL process, after uploading its model, the client hopes to receive more beneficial global information from the server to better assist in processing its local data. However, a challenge is that the client cannot determine whether the model received from the server contains more useful information for processing its local data. Liang et al. (2020)

and Collins et al. (2021) address this issue by decoupling the deep and shallow parameters of the model. Fallah et al. (2020) applies the MAML (Finn et al., 2017) framework in FL to construct an initialization model that performs well after a few rounds of updating on heterogeneous data. Hanzely & Richtárik (2020) propose constructing personalized models by combining global and local models. Sattler et al. (2020) clusters clients based on their similarity and performs federated learning within each cluster.

However, in these methods, clients do not directly verify the information contained in the models received from the server. For instance, if local data distributions significantly differ from the global model, the model may not generalize well, leading to poor performance. Additionally, in cases of adversarial or faulty clients, unverified models could be influenced by malicious updates, compromising both performance and security. In this case, for a resource-constrained client, which is quite commonly seen for internet of things devices, the client may encounter difficulties in performing multiple rounds of training to properly adjust the received aggregated model to make it suitable for the local heterogeneous data. As a result, the local model performance of these clients will not be guaranteed.

Therefore, we propose setting up a local supervisor to assist the model in fitting the local heterogeneous data using only a very limited number of communication rounds. Our proposed algorithm is termed **FedSimSup** (Federated learning with similarity information supervisor).

The contributions of our work are summarized as follows:

- We propose a novel supervisor-assisted pFL framework. Each client is assigned a local unique supervisor to monitor the information contained in the aggregated personalized model received from the server. If the information is beneficial to the client, the supervisor will update to improve supervision. Otherwise, the supervisor will guide the client to adjust the personalized model to be close to the state it was in after the last training.

- We propose leveraging the client's label similarity information to assist the model training via weighting personalized models. By evaluating relationships based on distribution differences of labels of different clients, each client can engage in selective learning from other clients. Through this selective learning process, clients can focus on integrating knowledge that is most applicable to their own context, improving overall model performance and efficiency.

The advantages of our proposed FedSimSup are as follows.

- We demonstrate its strong personalization capability, showing superior performance compared to other methods without the need for fine-tuning or other optimizations.

- Our method addresses the issue of model heterogeneity to some extent, allowing clients to build different model architectures based on their own needs and computational capabilities.

- Our method possesses a certain level of interpretability, enhancing clients' trust in the model and facilitating future exploratory research.

## 2 RELATED WORK

**Non-IID Data.** In real-world scenarios, Non-IID situation arises in various forms, such as attribute skew, label skew, temporal skew and data quality skew (Zhu et al., 2021). Among these, label skew is particularly prevalent and can significantly impact model performance. We focus primarily on label distribution skew, which can be categorized into label size imbalance and label distribution imbalance (Li et al., 2022). Label size imbalance (Pathological distribution) proposed in FedAvg (McMahan et al., 2017) firstly. In this setting, a hyperparameter $c$ is defined such that each user's dataset comprises data from only $c$ different categories, where a smaller $c$ indicates a more pronounced imbalance between clients. Label distribution imbalance (Dirichlet distribution) refers to the instances of labels for client $k$ following the distribution $p_{k,c} \sim Dir(\alpha)$, where $Dir(\cdot)$ represents the Dirichlet distribution (Hsu et al., 2019) and a smaller $\alpha$ indicates a greater degree of imbalance. In our work, we conducted experimental discussions on both types of label skew scenarios.

**Personalized Federated Learning** is an effective way to address data heterogeneous settings. Existing methods can generally be categorized into several types. First, *Data augmentation* (Jeong et al., 2018; Duan et al., 2019; Shin et al., 2020) aims to reduce data heterogeneity, enabling the use of the standard FL to address the problem. Following this, *Regularization* (Hanzely & Richtárik, 2020; T Dinh et al., 2020; Li et al., 2020) prevents client overfitting and accelerates global convergence, enhancing the overall robustness of the model. Additionally, *Meta learning* (Jiang et al., 2019; Fallah et al., 2020; Scott et al., 2024) enables the global model to achieve personalization more quickly on the client side. Furthermore, *Multi-task learning* (Smith et al., 2017; Huang et al., 2021) treats each client as a different task and leverage relationships between them to handle heterogeneous settings. Moreover, *Clustering* (Sattler et al., 2020; Briggs et al., 2020; Ghosh et al., 2020) divides clients into different homogeneous groups, whithin FL is performed more effectively. Lastly, *Knowledge distillation* (Li & Wang, 2019; Kamp et al., 2023) transfers knowledge from the server or other clients to a specific client, ensuring that each client benefits from shared insights.

*Parameter decoupling* refers to separating the model's parameters and implementing stepwise training, with one set of parameters being globally shared and another set trained locally, thereby enhancing the personalization capability. There are several main decoupling methods: The first method divides the network into base layers and personalized layers (Arivazhagan et al., 2019; Xu et al., 2023b; Liu et al., 2024), with the base layers being globally shared to obtain the generalized feature information, while the personalized layers are trained only locally to allow different clients to process the features in their own ways. The second method uses embeddings from the each client as personalization layers (Bui et al., 2019; Liang et al., 2020), aiming to extract unique features to be processed by the global model. Other methods, like Li et al. (2024) propose FedRAP which learns a global view and a personalized view locally on each client to achieve personalization. Parameter decoupling reduces the amount of transmitted parameters, thereby decreasing communication overhead to some extent. Although parameter decoupling has demonstrated its effectiveness in multiple aspects, it still faces challenges in handling scenarios with extreme data heterogeneity. Future research could explore more efficient decoupling strategies to optimize the performance of federated learning.

*Model interpolation* learns personalized models by combining local models with the global model, thus balancing the model's generalization and personalization capabilities. Hanzely & Richtárik (2020) designs a new objective function that incorporates a penalty term with a coefficient of $\lambda$. When $\lambda \to \infty$, it becomes FedAvg, and when $\lambda$ is zero, it corresponds to a model trained only locally. The value of $\lambda$ controls the trade-off between local and global differences. Additionally, Deng et al. (2020) propose a method to find an optimal combination of local and global models, aiming to enhance model performance under diverse client data distribution. Moreover, Chen et al. (2023) propose elastic aggregation, which performs adaptive interpolation based on the sensitivity of the model parameters, allowing for dynamic adjustments according to the specific needs of each client.

## 3 METHOD

### 3.1 PROBLEM FORMULATION

In this work, we assume supervised federated learning with a total of $n$ clients, each having its own Non-IID distributed dataset $D_i = \left\{ \left( x_1^i, y_1^i \right), \left( x_2^i, y_2^i \right) \cdots \left( x_{m_i}^i, y_{m_i}^i \right) \right\} \subset \mathcal{X} \times \mathcal{Y}$, for $i \in \{1, 2 \cdots n\}$, where $m_i$ is the amount of data for client $i$. We use both the Dirichlet method (Hsu et al., 2019) and the Pathological method (McMahan et al., 2017) to partition the data to simulate Non-IID distribution (Detailed partitioning methods are provided in the Appendix). Each client has a model (which may or may not be the same) $q_{\theta_i} : \mathcal{X} \to \mathcal{Y}$ maps input $x_j^i \in \mathcal{X}$ to predict label $q_{\theta_i} \left( x_j^i \right) \in \mathcal{Y}$ which is compared with the corresponding true label $y_j^i \in \mathcal{Y}$, $\theta_i \in \Theta$ represents the model parameters, and $\left( x_j^i, y_j^i \right)$ denotes one data in client $i$. The parameters of each client's model $\theta_i$ are trained based on its local dataset by minimizing the following objective function

$$\min_{\theta_i \in \Theta} L \left( D_i, \theta_i \right) = \frac{1}{m_i} \sum_{j=1}^{m_i} \ell \left( q_{\theta_i} \left( x_j^i \right), y_j^i \right), \quad (1)$$

where $\ell : \mathcal{Y} \times \mathcal{Y} \to \mathbb{R}_+$ is the loss function that measures the degree of inconsistency between the predicted labels $q_{\theta_i} \left( x_j^i \right)$ and true labels $y_j^i$.

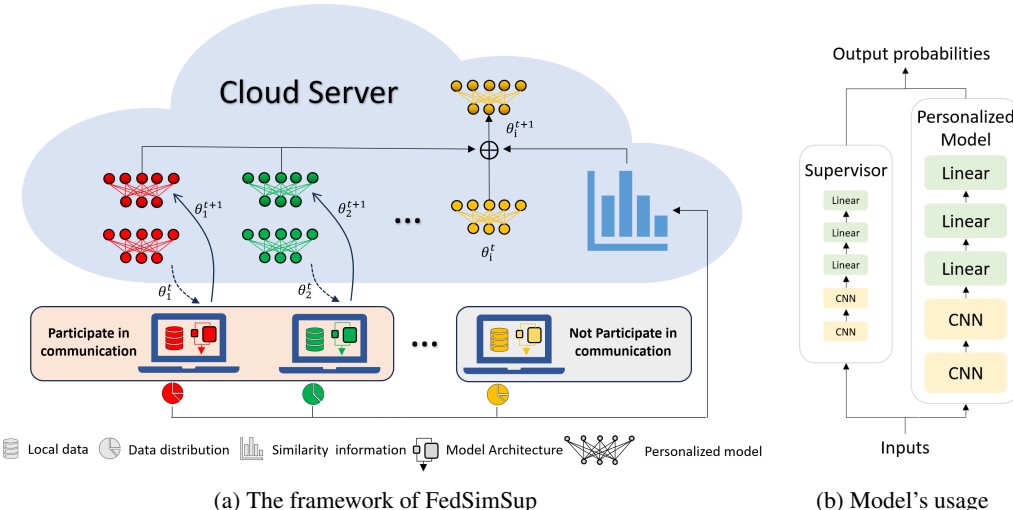

(a) The framework of FedSimSup      (b) Model's usage

Figure 1: (a) is the framework of FedSimSup. During each communication round, the server distributes the corresponding personalized model to the participating clients (red and green clients). These clients train their personalized models under the supervision of the local supervisor. Once the communication concludes, the personalized models are uploaded to the server, while the non-participating clients (yellow client) aggregate their personalized model with the trained personalized model based on similarity information. (b) shows the model architecture used by the client.

If each client has sufficient data and enough training resources, they can train a model that is suitable for their local data. However, this approach presents several issues: (1) In reality, not all clients have abundant data which severely affects the training of the model. (2) Some clients, such as those using portable devices like smartphones, may not support large-scale training (Pfeiffer et al., 2023). (3) When the model encounters data that it has not seen or has seen very little of during training, its performance will be poor (Zhu et al., 2021). To address these issues, federated learning has been proposed.

In standard FL, each client uses the same model, denoted as $q_{\theta_1} = q_{\theta_2} \cdots = q_{\theta_n}$, we refer to this model collectively as $q_\theta$. Let $\mathcal{N}(t)$ denotes the clients participating in the $t$-th communication round. The server distributes the model to these clients, who then train the model locally using the local objective function (1).

After training, the clients upload their models to the server for aggregation(McMahan et al., 2017):

$$\theta^{t+1} = \frac{\sum_{i \in \mathcal{N}(t)} (m_i \theta_i^t)}{\sum_{i \in \mathcal{N}(t)} m_i}, \tag{2}$$

where $\theta_i^t$ is the model of client $i$ after completing local training in the $t$-th communication round. This method takes into account the impact of data volume, aiming to allow clients with less data to learn from those with more data. However, this method performs poorly in terms of convergence speed and model performance in the presence of Non-IID data across clients. We propose FedSimSup in this work to address this issue.

## 3.2 LEARNING UNDER SUPERVISOR

In standard FL, one issue is that when the server sends the global model to local clients, the clients cannot determine whether the received model, containing global information, is more beneficial than the model trained in the previous round. To address this, we divide the model into two parts: the first part is the supervisor, which is trained locally but not uploaded. The second part is the personalized model, which is uploaded and aggregated. The reason why it has personalized characteristics will be explained in 3.3.

For demonstration purposes, we directly scale down the personalized model proportionally to create the supervisor, which then assists the personalized model in its usage. In practice, the architecture

of the supervisor does not need to be the same for every client. Each client can independently design their own supervisor architecture according to their specific needs and capabilities. The server only needs to manage the personalized model but not the whole local model. This approach significantly enhances the personalization capability of the model while simplifying management.We demonstrate the performance results when clients adopt different structures in 4.2. The structure of the model is shown in Figure 1b. And the local objective function also changes from (1) to

$$\min_{s_i \in \mathcal{S}, \theta_i \in \Theta} L\left(D_i, s_i, \theta_i\right) = \frac{1}{m_i} \sum_{j=1}^{m_i} \ell\left(q_{\theta_i}\left(x_j^i\right) + q_{s_i}\left(x_j^i\right), y_j^i\right), \tag{3}$$

where $s_i \in \mathcal{S}$ is the parameters of supervisor and $\theta_i \in \Theta$ is the parameters of personalized model. Here, we simply sum the results of the two models. The training process of our model is divided into two parts:

$$\min_{s_i \in \mathcal{S}} L\left(D_i, s_i, \theta_i\right), \tag{4}$$

$$\min_{\theta_i \in \Theta} L\left(D_i, s_i, \theta_i\right). \tag{5}$$

The purpose of (4) is that if the global model contains more beneficial information, the supervisor will undergo a significant update to better assist the training process. However, if the global model is not beneficial to the local data, the supervisor has already been fitted to the local data, we hypothesize that it will undergo only minor updates or remain unchanged. The purpose of (5) is to train the model under the supervision of the supervisor, ensuring that after acquiring global information, it becomes more fitted to the local data.

The role of the supervisor is to guide the local model during training by providing oversight based on the previously learned local data. It helps prevent the model from deviating too much from its fit to the local data while still incorporating beneficial global updates. The supervisor ensures that the personalized model maintains a balance between leveraging global information and staying aligned with the local distribution. We will demonstrate its supervisory assistance role in the 4.2.

### 3.3 UTILIZATION OF SIMILARITY INFORMATION

In the case of data heterogeneity, it is challenging to construct a model using (2) that performs well across all $n$ clients

$$\min_{\theta \in \Theta} \sum_{i=1}^{n} L\left(D_i, \theta\right), \tag{6}$$

where $\theta$ is the global model used by all clients. Therefore, we establish a personalized model for each client based on their local data distribution. At the end of a local training round, we perform the following operations on all clients' personalized models (the following operations are performed on the personalized model, unrelated to the supervisor).

If a client $i$ participates in this round of communication, then the personalized model $\theta_i^{t+1}$ of the $i$-th client at $t+1$ round is set to the updated $\theta_i^t$ after training without aggregating information from other clients

$$\theta_i^{t+1} = \theta_i^t, \qquad if \ \ i \in \mathcal{N}(t). \tag{7}$$

If the client $i$ does not participate in this round of communication, then $\theta_i^{t+1}$ of the $i$-th client is updated as follows.

$$\theta_i^{t+1} = \alpha_i^t \theta_i^t + \left(1 - \alpha_i^t\right) \sum_{j \in \mathcal{N}(t)} \frac{s_{ij}}{sum_i^t} \theta_j^t, \qquad if \ \ i \notin \mathcal{N}(t), \tag{8}$$

$$\alpha_i^t = \frac{K \cdot m_i}{\sum_{j \in \mathcal{N}(t)} m_j + K \cdot m_i}, \qquad sum_i^t = \sum_{j \in \mathcal{N}(t)} s_{ij}, \tag{9}$$

where $K$ is the number of clients participating in communication in each round, $\alpha_i^t$ is a parameter that measures the amount of data, calculated based on the ratio of the local data amount to the total data amount of clients participating in the $t$-th communication, which aligns with the original standard FL concept. $s_{ij} \in [0, 1]$ is the value that measures the similarity between client $i$ and client $j$. A larger value of $s_{ij}$ indicates a greater similarity between client $i$ and $j$. In (8), we aggregate the personalized models of clients who do not participate in communication, based on their data volume and the similarity between them and the clients actively participating in training. By doing this, we can ensure that clients that do not participate in training at each round can still benefit from the clients

---

**Algorithm 1** FedSimSup

---

**Input:** Dataset distributed across m clients $D = \{D_1, D_2 \cdots D_n\}$, client participating rate $r$, the number of global epochs $T$, personalized model epochs $\tau_\theta$, supervisor epochs $\tau_s$

1: Initialize $\theta_1^0, \theta_2^0 \cdots \theta_n^0, s_1^0, s_2^0 \cdots s_n^0$
2: **for** $t = 1, 2 \cdots T$ **do**
3:     $\mathcal{N}(t) \leftarrow$ server randomly samples $max(1, nr)$ clients
4:     **for** each client $i \in \mathcal{N}(t)$ **do**
5:         client $i$ initializes $s_i^{t,0} \leftarrow s_i^{t-1,\tau_s}$          ▷ Initialize the supervisor
6:         server sends $\theta_i^{t-1,\tau_\theta}$ to client $i$ as $\theta_i^{t,0}$          ▷ Initialize the personalized model
7:         $s_i^{t,\tau_s}, \theta_i^{t,\tau_\theta} \leftarrow$ **LocalUpdate**$(s_i^{t,0}, \theta_i^{t,0}, f_i, D_i)$          ▷ Train the two separately
8:         client $i$ sends updated personalized model $\theta_i^{t,end}$ to server
9:     **end for**
10:     **for** each client $i \notin \mathcal{N}(t)$ **do**
11:         set $s_i^{t,\tau_s} \leftarrow s_i^{t-1,\tau_s}$          ▷ The supervisor is not changed
12:         aggregate $\theta_i^{t,\tau_\theta}$ by (8)          ▷ Obtain global information based on similarity
13:     **end for**
14: **end for**
15:
16: **LocalUpdate**$(s^0, \theta^0, f, D)$**:**
17: **for** $j = 1, 2 \cdots \tau_s$ **do**
18:     $s^j \leftarrow SGD\left(f(s^{j-1}, \theta^0), s^{j-1}\right)$          ▷ Update the supervisor within $\tau_s$
19: **end for**
20: **for** $j = 1, 2 \cdots \tau_\theta$ **do**
21:     $\theta^j \leftarrow SGD\left(f(s^{\tau_s}, \theta^{j-1}), \theta^{j-1}\right)$          ▷ Update the personalized model within $\tau_\theta$
22: **end for**
23: **return** $s^{\tau_s}, \theta^{\tau_\theta}$

---

that participate in training, thereby promoting the effective global information aggregation. In this work, the similarity information is represented by the cosine similarity between the proportions of each client's data label distribution, which we believe better reflects the intrinsic similarity between clients. This requires us to collect the label proportions of cleints at the beginning of the entire task and compute the similarity between each client on the server, as shown in Figure 1a.

## 3.4 FEDSIMSUP ALGORITHM

We provide the pseudocode for FedSimSup in Algorithm 1, and below we will explain it in detail.

**Local Update** In each communication round, customers are randomly selected to participate based on a fixed participation rate $r$ and receive the personalized model $\theta$ sent by the server. Client $i$ participates in the $t$-th round, receives the personalized model $\theta_i^t$, and has a supervisor $s_i^t$ stored locally. The local supervisor is updated for $\tau_s$ epochs.

$$s_i^{t,j} \leftarrow SGD\left(f(s_i^{t,j-1}, \theta_i^{t,0}), s_i^{t,j-1}\right), \tag{10}$$

where $j \in (1, 2, \cdots \tau_s)$, and $\theta_i^{t,0}$ denotes the personalized model of client $i$ that has not been updated. we use Stochastic Gradient Descent (SGD) (Robbins & Monro, 1951) to update $s$ based on the gradient of $s$. Then, the personalized model is updated within round $\tau_\theta$:

$$\theta_i^{t,j} \leftarrow SGD\left(f(s_i^{t,\tau_s}, \theta_i^{t,j-1}), \theta_i^{t,j-1}\right), \tag{11}$$

where $j \in (1, 2, \cdots \tau_\theta)$. After completing these two processes locally, save the supervisor $s_i^{t,\tau_s}$ and upload the personalized model $\theta_i^{t,\tau_\theta}$ for aggregation of other clients.

**Server Update** The server receives the personalized models uploaded from client set $\mathcal{N}(t)$, without modifying them. For clients who did not participate in the communication, it aggregates their models based on (2), leveraging similarity information to learn from the clients that have participated in this round of training.

## 4 EXPERIMENTS

### 4.1 EXPERIMENTAL SETTINGS

**Datasets** We evaluate FedSimSup by classification tasks using the CIFAR10, CIFAR100 (Krizhevsky et al., 2009) and FEMNIST (Caldas et al., 2018). CIFAR10 and CIFAR100 are among the most classic image classification tasks, both containing $60,000$ images, evenly distributed across 10 and 100 categories, respectively. We let each client follow a Dirichlet distribution with $\alpha$ values of 0.1 and 0.5 to simulate a Non-IID setting for CIFAR10 and CIFAR100 datasets. FEMNIST is a dataset with 62 different character categories (including numbers and uppercase and lowercase English letters), with a total of 805,263 samples. We test the performance of our proposed FedSimSup and algorithms under comparison under the Dirichlet distribution for CIFAR10, CIFAR100, and FEMNIST datasets. We also test the performance under the Pathological distribution for CIFAR10 and CIFAR100 datasets. Details of data partitioning are given in the Appendix.

**Baselines** We compare FedSimSup with six methods, including FedAvg (McMahan et al., 2017), Per-FedAvg (Fallah et al., 2020), FedRep (Collins et al., 2021), FedProto (Tan et al., 2022b), Fed-Prox (Li et al., 2020) and FedPac (Xu et al., 2023a). In FedProx, a proximal term is used to improve stability. Per-FedAvg proposes using the MAML framework to obtain an initial model that quickly adapts to clients. FedRep (Collins et al., 2021) sets up a unique head for each client to enhance personalization capability. FedProto (Tan et al., 2022b) aggregates the local prototypes to avoid gradient misalignment. FedPac (Xu et al., 2023a) conducts explicit local-global feature alignment by leveraging global semantic knowledge. Additionally, we also compare our FedSimSup with the performance of conducting local training separately on each client.

**Settings for Baselines** During local training, we also randomly select clients at a proportional rate in each round and conduct training, but we do not perform aggregation. This means that the client's model will only change after client participates in communication. In the FedAvg method, we set the client participation rate to 0.1, the number of communication rounds to 1000, and the local epochs to 5. For other methods, unless specified otherwise, the parameters remain the same. In the FedProx method, we set the $\mu$ to 1 to improve stability. In the FedPac method, we set $\lambda$ to 1. In the Per-FedAvg method, we set $\tau$ to 4 and $\alpha$ to 0.001, and use Per-FedAvg (HF). During testing, each client performs fine-tuning for 3 epochs. In the FedRep method, we set the classification head as the personalized layer, training the classification head for 2 epochs and the representation layer for 3 epochs. In the FedProto method, we set the importance weight $\lambda$ to 1.

**Model** Like most pFL approaches, FedSimSup uses the LeNet-5 (LeCun et al., 1998) as the local model for each client, considering the communication cost. LeNet-5 consists of two convolutional layers and two linear layers. For fairness, we use LeNet-5 as the model for all algorithms under comparison in this work. Since our FedSimSup includes both a supervisor and a personalized model in each client. Thus, to ensure the number of parameters of FedSimSup is almost same as that of competing algorithms, we proportionally reduce the size of LeNet-5 to approximately one-sixth of that of the personalized model. In the experiments, to test the influence of different supervisor architectures on the performance, we let each client randomly select one from three types of architectures, i.e., the aforementioned LeNet-5, a smaller convolutional neural network (CNN), and a large transformer structure (Vaswani, 2017), to simulate real-world client scenarios. These three different models represent the differences in computational capabilities, needs, and intellectual property among clients in the real world.

**Training Details** We set the global communication rounds to 1,000 and the local training epochs to 5, with 3 epochs dedicated to training the personalized model and 2 epochs for training the supervisor. For CIFAR10 and CIFAR100, we set the number of clients to 50 and 100 with a participation rate of 0.1 per round. For FEMNIST we maintain its original setup with a total of 3,597 clients to ensure that our method remains effective under a large number of clients. About the participation, we set it to 0.1 for local training and 0.01 for other methods. We set the batch size for SGD to 32 and the learning rate to 0.1. The detailed settings for other methods will be mentioned in the Appendix.

### 4.2 EXPERIMENTAL RESULTS

Table 1 compares FedSimSup with other methods under the Dirichlet distribution, showing that FedSimSup is optimal in all tested cases. Notably, on the more challenging CIFAR100 dataset,

Table 1: Accuracy under Dirichlet distribution (best valued per setup in bold).

| clients num(Dir) | CIFAR10 | | | | CIFAR100 | | | | FEMNIST |
| | 100(0.1) | 50(0.1) | 100(0.5) | 50(0.5) | 100(0.1) | 50(0.1) | 100(0.5) | 50(0.5) | 3597 |
|---|---|---|---|---|---|---|---|---|---|
| Local | 86.67 | 86.3 | 59.4 | 61.81 | 40.44 | 43.28 | 17.99 | 21.6 | 66 |
| FedAvg | 33.45 | 43.22 | 50.89 | 54.91 | 20.2 | 20.89 | 23.34 | 27.01 | 79.76 |
| FedProx | 33.24 | 37.9 | 51.18 | 54.87 | 19.43 | 19.89 | 22.36 | 26.1 | 74.2 |
| Per-FedAvg | 79.12 | 79.09 | 38.13 | 50.44 | 3.92 | 10.69 | 1.59 | 3.11 | 2.57 |
| FedRep | 88.43 | 88.18 | 71.96 | 73.89 | 46.48 | 52.03 | 25.8 | 32.59 | 81.26 |
| FedProto | 86.75 | 86.15 | 59.98 | 61.85 | 41.55 | 43.61 | 17.61 | 22.13 | 9.98 |
| FedPac | 86.41 | 85.59 | 66.59 | 68.15 | 41.23 | 43.52 | 23.2 | 23.97 | 78.24 |
| FedSimSup | **89.73** | **88.88** | **73.9** | **75.08** | **50.67** | **55.48** | **32.5** | **39.23** | **84.32** |

Table 2: Accuracy under Pathological distribution (best valued per setup in bold).

| clients num(Shard) | CIFAR10 | | | | CIFAR100 | | | |
| | 100 (2) | 50 (2) | 100 (5) | 50 (5) | 100 (5) | 50 (5) | 100 (20) | 50 (20) |
|---|---|---|---|---|---|---|---|---|
| Local | 86.07 | 88.3 | 65.2 | 68.4 | **66.72** | **67.32** | 27.93 | 34.98 |
| FedAvg | 40.15 | 39.13 | 51.8 | 53.41 | 12.97 | 14.98 | 20.21 | 21.65 |
| FedProx | 38.96 | 35.63 | 51.71 | 53.02 | 12.52 | 13.79 | 19.51 | 21.17 |
| Per-FedAvg | 51.59 | 70.67 | 29.1 | 51.43 | 2.97 | 9.34 | 1.43 | 5.24 |
| FedRep | 86.65 | **88.57** | 74.52 | **77.24** | 62.96 | 67.27 | 39.4 | 46.4 |
| FedProto | 86.09 | 87.76 | 64.04 | 67.31 | 65.83 | 66.33 | 28.19 | 34.09 |
| FedPac | 85.48 | 87.67 | 71.27 | 72.68 | 54 | 59.92 | 21.29 | 34.53 |
| FedSimSup | **87** | 88.07 | **75.75** | 76.99 | 63.91 | 65.77 | **43.83** | **48.87** |

it demonstrates an improvement of about 4 - 6% compared to the second-best method. Table 2 presents the experimental comparison of FedSimSup under the Pathological distribution, where it can be seen that FedSimSup is not always the best method. Upon analysis, we believe this is due to the similarity computation under the Pathological distribution resulting in only a few possible discrete values, which affects the finer differentiation of similarity between clients, thus leading to a performance that is not as good as that under the Dirichlet distribution.

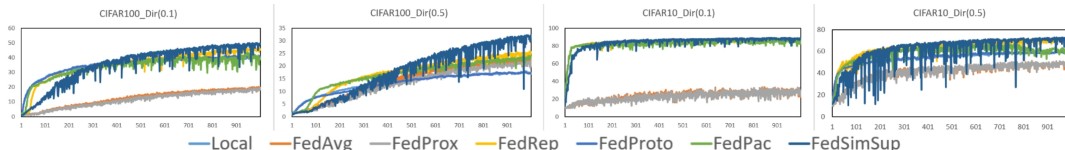

Figure 2: Comparison of convergence speeds among different methods.

**Convergence Analysis** We found that using similarity information accelerates convergence speed information accelerates convergence speed, which has practical significance in cases with limited communication. In Figure 3, we compare the impact of using similarity information versus not using it on convergence speed. The experiment has been conducted on CIFAR10, and we display the results for the first 100 epochs. Results show that the use of similarity information do accelerate convergence speed, demonstrating the effectiveness of our proposed similarity measurement and aggregation strategy.

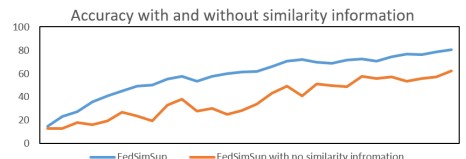

Figure 3: Comparison of convergence speed with and without similarity information.

We also compared the convergence speeds of different methods. In Figure 2, the accuracy changes of various methods under non-iid distributions of CIFAR-10 and CIFAR-100, with Dir(0.1) and Dir(0.5), over 1000 epochs are presented. In the more challenging CIFAR-100 task with a larger number of categories, our method shows a slower initial improvement. However, by learning from other clients based on similarity, it can acquire knowledge that is more akin to its own, leading to better overall performance. Furthermore, our method exhibits larger fluctuations in performance under the Dir(0.5) distribution. We believe this is due to the

Table 3: Experiments of using different supervisor architectures.

| | CIFAR10 | | | | CIFAR100 | | | |
|---|---|---|---|---|---|---|---|---|
| clients num(Dir) | 100 (0.1) | 50 (0.1) | 100 (0.5) | 50 (0.5) | 100 (0.1) | 50 (0.1) | 100 (0.5) | 50 (0.5) |
| FedSimSup-T | **90.77** | **91.27** | 70.45 | 73.05 | **51.32** | **55.73** | 30.14 | 36.12 |
| FedSimSup-C | 84.49 | 87.9 | 69.91 | 71.85 | 44.32 | 44.95 | 25.63 | 30.77 |
| FedSimSup-L | 86.19 | 85.38 | 67.77 | 70.21 | 45.28 | 47.51 | 23.44 | 32.24 |
| Whole | 87.36 | 88.24 | 69.41 | 71.73 | 46.82 | 50.23 | 26.72 | 32.95 |
| Original | 89.73 | 88.88 | **73.9** | **75.08** | 50.67 | 55.48 | **32.5** | **39.23** |

relatively small differences between clients, making them less sensitive to variations in similarity. Therefore, our method tends to achieve better results on tasks that are more challenging and have greater disparities.

**Supervisory Assistance** We verify the supervisory assistance effect of the supervisor using Class Activation Map (CAM) (Selvaraju et al., 2017) in image classification tasks. As shown in Figure 4, the image on the left is the original classification task image, the middle one is the CAM of the pe-

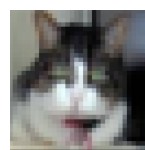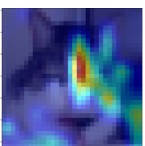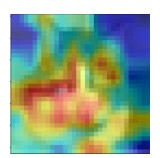

rsonalized model, and the one on the right is the CAM of the supervisor. It can be observed that, when trying to recognize the image as a cat, the personalized model, possibly influenced by information learned from other clients, tends to focus on scattered details, such as the cat's eyes or nose. In contrast, the super-visor focuses on the entire body of the cat, helping to prevent the personalized model's attention from devi-

Figure 4: CAM of the personalized model (middle) and the supervisor (right).

ating too much. Thus, we conclude that the supervisor and personalized model in our FedSimSup have different focuses, enhancing the interpretability of the model's behavior.

**Different Supervisor Architectures** We simulate three types of clients employing different supervisor architectures to observe their effects. These include a transformer architecture with a larger number of parameters (FedSimSup-T), a CNN network with fewer parameters (FedSimSup-C), and the original small LeNet-5 architecture (FedSimSup-L). Table 3 shows the performance of clients using these different supervisor architectures in the same federated learning process. "Whole" represents the combined performance of the three types of clients, while "Original" shows the performance of the original method where all clients used the same LeNet-5 supervisor architecture. As observed in Table 3, only adopting the transformer architecture shows better performance in the two CIFAR datasets than FedSimSup-C and FedSimSup-L. This is reasonable since the transformer architecture has the largest number of parameters. We also observe that the overall performance with different ar-chitectures (row 4 in Table 3) is slightly worse compared to when all clients use the same supervisor architecture (row 5 in Table 3). This is an unavoidable consequence of model heterogeneity. Despite this, the performance gap is not large, and some clients achieved better results by selecting models that fit their individual needs. Therefore, we conclude that our proposed FedSimSup is flexible to include different model architectures for different clients according to their computational resources and needs, allowing them to achieve better performance and faster inference.

## 5 CONCLUSION AND FUTURE WORK

In this work, to address the issue in federated learning where the global information sometimes de-viates too much from the local data and clients learn indiscriminately from other clients, we propose a novel pFL method, FedSimSup. Our approach allows each client to employ their own supervisor with flexible architectures to assist local training, preventing the model from deviating too much from the local data. Additionally, we utilize the similarity information to standardize the way of clients learning from other clients' information. Overall, FedSimSup provides better performance in handling Non-IID scenarios, while allowing clients the freedom to customize their model architec-tures and offering a certain level of interpretability. In FedSimSup, our similarity measurement only considers differences in distribution of labels, resulting in slightly worse performance on patholog-ically distributed data. Also, the similarity information remains static, but during the learning pro-cess, a deeper understanding of the similarity between clients should be more helpful for improving

the overall performance. Thus, one of our future work will focus on designing dynamic similarity measurements to handle various label distributions. Additionally, since our proposed FedSimSup can accommodate different model architectures for different clients, another direction of our future work will focusing on studying what the most effective combination of model architectures for all clients to simultaneously balance the overall algorithm performance and clients' own computational ability.

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

## A APPENDIX-EXPERIMENTS

### A.1 DATA PARTITIONING

Our data partitioning only considers the label differences between clients.

**Pathological Non-IID partition** In pathological distribution, we first need to determine the number of categories $c$ to be distributed to each client. We will partition the data based on the total amount of data, the number of categories, the number of clients, ensuring that each piece of data does not appear more than once and that all data is utilized. We present our partitioning on CIFAR-10, as shown in Figure 5a.

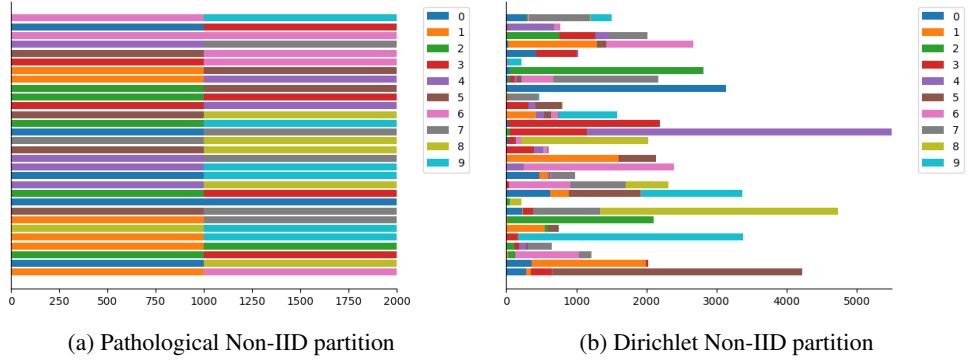

(a) Pathological Non-IID partition      (b) Dirichlet Non-IID partition

Figure 5: Partitioning on CIFAR-10

**Dirichlet Non-IID partition** In the Dirichlet distribution, the distribution for each client is independent. Assume that the distribution for client is governed by a vector $q$ $(q_i > 0, i \in [1, M], \|q\|_1 = 1)$ of length $M$, where $M$ represents the number of classes. The vector $q$ is sampled from a Dirichlet distribution

$$q \sim Dir(\alpha p) \tag{12}$$

$p\left(p_i > 0, i \in [1, M], \|p\|_1 = 1\right)$ represents the prior class distribution that we manually set. Here, we define them as $p_i = \frac{1}{M}, i \in [1, M]$. The parameter $\alpha$ is a concentration parameter, which can be simply understood as determining the probability that a sample belongs to the prior $p$ When each element in $p$ is the same. The probability density function of the Dirichlet distribution is given by:

$$(q \mid \alpha p) = \frac{1}{B\left(\alpha p\right)} \prod_{i=1}^{M} q_i^{\alpha p_i - 1} \tag{13}$$

$$B\left(\alpha p\right) = \frac{\prod_{i=1}^{M} \Gamma\left(\alpha p_i\right)}{\Gamma\left(\sum_{i=1}^{M} \alpha p_i\right)} \tag{14}$$

And $E\left(q_i\right) = p_i$. We can see from 13 that when $\alpha p_i$ is large, our samples are nearly $q_i = \frac{1}{M}, i \in [1, M]$, whereas when $\alpha p_i$ is small, only one category appears in the samples. Therefore, we can set the size of $\alpha p$ to control the degree of Non-IID data. Since each element in $p$ is the same and we are only concerned with the size of $\alpha p$, we can set just one variable $\alpha$ to automatically normalize $p$ and control the generation of the desired data.

However, this partitioning method still presents some issues. First, different clients may have overlapping data, or certain data in the dataset may not be utilized. Second, the number of samples for each client is predetermined and the same across all clients, which is almost impossible in real-world scenarios because clients vary in their ability to collect data. Therefore, we apply the Dirichlet distribution to the data for each class, where $q$ and $p$ become vectors of size $N$, where $N$ is the number of clients. During the partitioning process, we need to ensure that a larger portion of the data is allocated to clients with fewer overall data points to maintain a Non-IID distribution. However, a problem arises when there are too many clients: insufficient data may result in some clients having too little data after all categories have been split. In this case, we can repartition the data until the client with the least amount of data reaches the required threshold. We present our partitioning on CIFAR-10, as shown in Figure 5b.

