# OpenReview forum: "Personalized Federated Learning With Similarity Information Supervisor"
_ICLR.cc/2025/Conference — Submitted to ICLR 2025_

### Official Review · Reviewer_r3xw · 2024-10-24

**Soundness:** 2
**Presentation:** 2
**Contribution:** 2
**Rating:** 3
**Confidence:** 3

**Summary:**

The paper addresses the challenge of data heterogeneity in federated learning, which can lead to model weight divergence and performance deterioration. This paper proposes a novel personalized federated learning (pFL) method called Federated Learning with Similarity Information Supervision (FedSimSup). The key idea behind FedSimSup is to integrate a local supervisor that refines the personalized model when the broadcast global model is not beneficial to local model performance.

**Strengths:**

1.	The paper makes an original contribution by proposing a novel hierarchical client clustering approach for federated learning, effectively addressing the challenge of data heterogeneity across clients.
2.	The paper is reasonably clear in structure and presentation, with a logical flow that generally allows readers to follow the authors' approach.

**Weaknesses:**

1.	While the paper attempts to introduce a hierarchical client clustering method in the field of federated learning, the level of innovation is relatively limited. The concept of client clustering has been explored in existing research, it lacks breakthrough novelty.

2.	The experimental section lacks sufficient comparison experiments and diverse datasets, making it difficult to convincingly demonstrate the effectiveness of the proposed method across different scenarios. The description of the client clustering strategy in federated learning is detailed, but the depth and breadth of the experiments are insufficient.

3.	Certain technical details are described ambiguously, particularly in the client clustering process and the complexity analysis of the algorithm. This could make it difficult for readers to fully understand the key points of innovation.

4.	Figure 2 and Figure 4 are not visually clear, which hinders the reader’s ability to interpret the results effectively. The lack of clear visualizations and detailed performance analysis further weakens the overall quality.

5.	The paper lacks a strong discussion of future research directions, failing to show the potential long-term impact of its approach.

**Questions:**

1.	Both Figure 2 and Figure 4 are not visually clear, making it difficult to interpret the experimental results effectively. Could you provide improved versions of these figures with clearer labels, higher resolution, and better color differentiation?

2.	The current experiments are limited to a small set of datasets and lack comparisons with more recent or diverse methods in the field. Could you extend the experimental evaluation to include a wider variety of datasets and compare the proposed method against more recent federated learning techniques that address data heterogeneity?

3.	Could you expand on the possible challenges or limitations of your method in more complex federated learning scenarios? Additionally, what future improvements or directions do you envision for enhancing the clustering process or further addressing data heterogeneity?

---

### Official Review · Reviewer_xEpY · 2024-10-30

**Soundness:** 2
**Presentation:** 2
**Contribution:** 2
**Rating:** 3
**Confidence:** 4

**Summary:**

This paper proposed a novel personalized federated learning method called FedSimSup, which addresses the issue of data heterogeneity by introducing local supervisors to assist in model training and combining personalized models for global information aggregation. The method considers the similarity relationships among clients and enhances information utilization efficiency through weighted personalized models. Experimental results show that FedSimSup outperforms seven state-of-the-art federated learning methods when handling heterogeneous data.

**Strengths:**

1) The FedSimSup method introduces the concept of local supervisors into traditional personalized federated learning, providing a new approach to address the issue of data heterogeneity.
2) The baseline methods are diverse: through comparisons with various federated learning approaches, FedSimSup demonstrates strong performance.
3) FedSimSup demonstrates a certain level of interpretability in its information utilization.

**Weaknesses:**

1)The theoretical support of the paper is insufficient. Although the FedSimSup method is proposed, there is a lack of in-depth analysis of its theoretical foundations. For example, key issues such as how to ensure the effectiveness and stability of the similarity measurement, as well as the specific impact of the supervisor model on the final model's performance, are not adequately addressed.
2)In Section 3.2, to address the issue that "clients cannot determine whether the received model (containing global information) is more beneficial than the model trained in the previous round," the authors divide the model into a supervisor and a personalized part. However, it is unclear why, after this division, clients can assess whether the received information is more advantageous. What metrics are used to quantify this judgment? The authors should provide necessary explanations.
3)In Section 4.2, the performance of FedSimSup under the Pathological distribution is poor. The authors attribute this to the limited discrete values resulting from similarity calculations under the pathological distribution, which affects the differentiation of similarity between clients. This explanation lacks theoretical and experimental support. It is recommended that the authors conduct a more in-depth discussion and analysis.
4)The specific implementation of similarity measurement is unclear. The paper mentions using cosine similarity to calculate the similarity between clients, it lacks detailed explanations on how to specifically implement and optimize this process. There is also no discussion on how to maintain the stability and accuracy of similarity calculations under different label distributions.
5)The connections between various components are still unclear, and there is a lack of ablation studies. The paper does not conduct ablation experiments to verify the impact of different components (such as the separation of the supervisor and personalized model) on overall performance, making it impossible to determine the importance and necessity of each part.
6)The complexity of the model has not been adequately considered. The paper mentions that different clients can use different supervisor architectures, it does not provide specific guidance or standards for selecting these architectures. Additionally, it does not discuss the impact of model complexity on training time and resource consumption.
7)The evaluation of privacy performance has not been considered. Although the paper discusses personalized learning and model aggregation, there is limited consideration of privacy protection. It lacks a discussion on how to effectively utilize similarity information while ensuring client privacy.

**Questions:**

See the weakness.

---

### Official Review · Reviewer_dbQQ · 2024-10-31

**Soundness:** 2
**Presentation:** 1
**Contribution:** 2
**Rating:** 3
**Confidence:** 4

**Summary:**

This paper a novel personalized federated learning (pFL) method called Federated Learning with Similarity Information Supervision (FedSimSup) to address the challenge of data heterogeneity across clients in federated learning.FedSimSup integrates a local supervisor mechanism and a personalized model to facilitate effective global information aggregation while respecting the unique data distributions of individual clients.

**Strengths:**

1) The supervisor module demonstrates a certain degree of flexibility, enabling model heterogeneity at a relatively low cost.
2) The paper presents model interpretability in a straightforward and intuitive manner.

**Weaknesses:**

1) In Section 3.3, what is the rationale behind not aggregating other clients' data for client i? Additionally, Equation (8) describes the calculation of similarity among all clients to create and deploy different models. Is this calculation performed on the server or client side? If on the client side, there may be issues with security and communication overhead; if on the server side, there could be computational overhead for generating different models. The author is requested to provide a detailed response to this issue.

2) The method description is overly concise. For instance, the exact similarity calculation method and how the supervisor guides the model are not elaborated. If the method merely stitches together similarity computation and the supervisor module, the paper may lack sufficient contribution.

3) In Table 1, what does the 3597 value for FEMNIST represent? Additionally, Figure 6 is too small, and the significant fluctuations in the proposed method's performance suggest potential design issues. Could there be an algorithmic problem contributing to this?

4) Numerous recent methods address parameter decoupling, such as FedCP, which extracts both global and local information from data. Its extraction modules are aggregable. Compared to this, the lack of aggregation in the supervisor module here might lead to overfitting. The author is advised to conduct a thorough survey of recent pFL methods and benchmark against newer approaches.

5) If the supervisor module is only conducting gradient descent, would increasing the parameter size of the local model achieve similar results without this module? It is recommended to add ablation experiments and consider this question in further detail.

**Questions:**

See the above comments.

---

### Official Review · Reviewer_AY6u · 2024-11-03

**Soundness:** 2
**Presentation:** 2
**Contribution:** 2
**Rating:** 3
**Confidence:** 4

**Summary:**

This paper introduces FedSimSup, a novel pFL method that uses a local supervisor to refine the personalized model and ensure effective global information aggregation. FedSimSup also measures client similarity based on label distribution differences to enhance information sharing among similar clients. Experimental results show that FedSimSup outperforms seven state-of-the-art methods on heterogeneous data, supports different model architectures across clients, and offers interpretability.

**Strengths:**

1. The paper introduces FedSimSup, a new method in personalized federated learning that incorporates a local supervisor to refine personalized models.
2. FedSimSup allows for different model architectures across clients, making it adaptable to various computational capacities and needs.
3. Experimental results show that FedSimSup outperforms seven state-of-the-art federated learning methods on heterogeneous data, demonstrating strong capabilities.

**Weaknesses:**

1. From Equations (3), (4), and (5), it appears that the client models are training two models simultaneously. This does not seem to follow a clear supervision process like knowledge distillation. Could the authors provide a clearer explanation of how the supervision mechanism operates within this framework?

2. The paper mentions using label distribution differences to compute $s_{ij}$. It would be helpful if the authors could elaborate on the exact method used to evaluate this similarity, ensuring clarity on its computation based on label distributions.

3. Is it possible to observe the effects of the similarity metric $s_{i,j}$ during the training process? Specifically, for a given client $i$, could the authors illustrate how it determines which other clients it tends to select throughout the training?

4. Could the authors provide the memory usage of FedSimSup compared to other baseline methods, particularly when different model architectures are employed on the client side?

5. In Table 3, FedSimSup-T seems to underperform compared to the original method utilizing LeNet-5, despite having more parameters in the Transformer model. Could the authors provide any insights for this case?

**Questions:**

Please see the weaknesses.

---

### Official Review · Reviewer_psBQ · 2024-11-03

**Soundness:** 2
**Presentation:** 2
**Contribution:** 1
**Rating:** 3
**Confidence:** 5

**Summary:**

This paper presents a personalized federated learning method named FedSimSup, designed to address the challenges posed by heterogeneous (Non-IID) data across clients in federated learning. Traditional federated learning methods may struggle with model weight divergence when clients have diverse data distributions, degrading performance. FedSimSup aims to tackle this challenge by introducing a local supervisor for each client to ensure the personalized model remains aligned with local data. This supervisor can override global updates if they are not beneficial, effectively balancing the integration of global and local knowledge.

Key contributions include the following ones.
1. FedSimSup assigns a unique local supervisor to each client, allowing the model to selectively integrate global updates based on their relevance to the client's data. If an update isn't beneficial, the supervisor maintains the client model's previous state, enhancing performance with minimal communication rounds.
2. FedSimSup uses label distribution similarity between clients to enable selective information sharing, allowing clients to learn primarily from those with similar data distributions, improving model personalization.
3. It supports various model architectures across clients, adapting to the specific computational capabilities and needs of each client. The model also provides interpretability through Class Activation Maps that visualize the supervisor's influence in keeping model attention aligned with relevant features.

**Strengths:**

See the contributions in the summary.

Also, two highlights of strengths:
1. Information among similar clients are utilized.
2. local model aggregation is somehow effective.

**Weaknesses:**

1. Insufficient Justification for Additional Model: It's needed to clarify the necessity of adding a local supervisor model that frequently participates in training. Note that there are many approaches to avoid adding a trainable model for each client, which would significantly increase resource consumption in terms of space and time.

2. Unclear Model Heterogeneity: This work mentions that the proposed method enables model heterogeneity to accommodate different mobile devices. However, it appears that heterogeneity is only considered for the supervisor model which may not be necessary. Typically, model heterogeneity applies to the client's primary model (i.e., the personalized model) or to the entire client framework. Since this work focuses on the supervisor model alone, the contribution regarding model heterogeneity is unclear and not well-justified.

3. Unclear contributions regarding explainability: This work uses CAM to demonstrate only explainability rather than interpretability. Adding external XAI mechanisms to enhance explainability is not an inherent advantage of the proposed method itself.

4. Unclear Data Heterogeneity: The paper uses Dirichlet and pathological distributions but should clarify which specific type of heterogeneity the method targets, as data heterogeneity includes many categories. Besides, even within Dirichlet and pathological distributions, different parameter settings have distinct real-world implications, which should be specified in the introduction.

5. Methodology Clarity: The motivation behind the supervisor's role is unclear, as is the explanation of similarity information. Figure 1(a) could be confusing, particularly regarding the communication rounds.

6. Lack of Theoretical Analysis: Although convergence is demonstrated experimentally, this paper in the present form lacks theoretical analysis of the method.

7. Some issues in experiments and results:
7.1 Unconvincing Baseline Selection: The baseline choices are either insufficiently targeted or poorly explained. Baselines with similar designs or research objectives should be selected for comparison.
7.2 Lack of SOTA Performance: As shown in Figures 1 and 2, the proposed method is not always optimal on pathological distributions, and its advantage on Dirichlet distributions is not always prominent.
7.3 The results shown in Tables 1 and 2 need detailed explanation. Please also explain why Per-FedAvg performs so poorly.
7.4 Given the addition of a supervisor, the experiments should include an analysis of time and space complexity.
7.5 Lack of Privacy Protection Discussion: Since model is communicated, it would be necessary to provide some discussion of privacy preserving.

**Questions:**

Please refer to weaknesses.

---

### Meta-Review · Area_Chair_mwh6 · 2024-12-20

**Metareview:**

The paper received five negative ratings, with all reviewers inclined to reject it. While presenting a promising approach, it has significant weaknesses. It introduces FedSimSup, which aims to enhance federated learning with a supervisor model, but the justification for adding a local supervisor model is unclear, and the application of model heterogeneity is insufficiently explained. Contributions on explainability are limited to using Class Activation Mapping (CAM), without providing inherent interpretability. Data heterogeneity is not well-defined, and the paper does not clarify the types of heterogeneity addressed or their real-world implications. The calculation of similarity lacks detail, leaving uncertainties about its implementation. The paper also lacks a theoretical analysis to support experimental results, such as the stability of the similarity measure and the supervisor's impact on performance. The experimental design is weak, with poor baseline selection, insufficient performance on pathological distributions, and no analysis of time or space complexity. Privacy protection in model communication is not addressed. The paper would benefit from clearer visualizations, detailed ablation studies, and better performance evaluations across diverse datasets. While the results show promise, they lack consistent superiority, and the rationale behind some design choices remains unclear. The authors do not adequately discuss future research directions. Given these issues and the lack of response from the authors, the Area Chair recommends rejection.

**Additional Comments On Reviewer Discussion:**

The paper received five negative ratings, with all reviewers inclined to reject it.

---

### Decision · Program_Chairs · 2025-01-22

Reject